# Association of Amlodipine with the Risk of In-Hospital Death in Patients with COVID-19 and Hypertension: A Reanalysis on 184 COVID-19 Patients with Hypertension

**DOI:** 10.3390/ph15030380

**Published:** 2022-03-21

**Authors:** Gwenolé Loas, Philippe Van de Borne, Gil Darquennes, Pascal Le Corre

**Affiliations:** 1Department of Psychiatry, Hôpital Erasme, Université Libre de Bruxelles (ULB), 1070 Brussels, Belgium; gil.darquennes@erasme.ulb.ac.be; 2Research Unit (ULB 266), Hôpital Erasme, Université Libre de Bruxelles (ULB), 1070 Brussels, Belgium; 3Department of Cardiology, Hôpital Erasme, Université Libre de Bruxelles (ULB), 1070 Brussels, Belgium; philippe.van.de.borne@erasme.ulb.ac.be; 4Pôle Pharmacie, Service Hospitalo-Universitaire de Pharmacie, CHU de Rennes, 35033 Rennes, France; pascal.le-corre@univ-rennes1.fr; 5Irset (Institut de Recherche en Santé, Environnement et Travail)-UMR_S 1085, CHU Rennes, INSERM, EHESP, University of Rennes, 35000 Rennes, France; 6Laboratoire de Biopharmacie et Pharmacie Clinique, Faculté de Pharmacie, Université de Rennes 1, 35043 Rennes, France

**Keywords:** functional inhibitors of acid sphingomyelinase (FIASMAs), COVID-19, SARS-CoV-2, mortality, amlodipine, calcium channel blockers (CCBs), calcium

## Abstract

Association between calcium channel blockers (CCBs) or functional inhibitors of acid sphingomyelinase (FIASMAs) use and decreased mortality in people with COVID-19 has been reported in recent studies. Since amlodipine is both a CCB and a FIASMA, the aim of this study was to investigate the association between chronic amlodipine use and the survival of people with hypertension infected with COVID-19. This retrospective cohort study used data extracted from the medical records of adult inpatients with hypertension and laboratory-confirmed COVID-19 between 1 March 2020 and 31 August 2020 with definite outcomes (discharged from hospital or deceased) from Erasme Hospital (Brussels, Belgium). We re-analyzed the data of the retrospective cohort study using only the 184 patients (103 males, 81 females) with a mean age of 69.54 years (SD = 14.6) with hypertension. The fifty-five participants (29.9%) receiving a chronic prescription of amlodipine were compared with the 129 patients who did not receive a chronic prescription of amlodipine. Univariate and multivariate logistic regressions were used to explore the relationships between mortality and sex, age, comorbidities, smoking, and amlodipine status. Out of the 184 participants, 132 (71.7%) survived and 52 (28.3%) died. The mortality rates were, respectively, 12.73% (*n* = 7) and 34.88% (*n* = 45) for the amlodipine and non-amlodipine groups. Multivariate logistic regression was significant (Chi square (5) = 29.11; *p* < 0.0001). Chronic kidney disease and malignant neoplasm were significant predictors as well as amlodipine status. For chronic kidney disease and malignant neoplasm, the odds ratio with 95% confidence interval (95% CI) were, respectively, 2.16 (95% CI: 1.04–4.5; *p* = 0.039) and 2.46 (95% CI: 1.01–6.01; *p* = 0.047). For amlodipine status the odds ratio was 0.29 (95% CI: 0.11–0.74; *p* = 0.009). The result of the present study suggests that amlodipine may be associated with reduced mortality in people with hypertension infected with COVID-19. Further research and randomized clinical trials are needed to confirm the potential protective effect of amlodipine in people with hypertension infected with COVID-19.

## 1. Introduction

Coronavirus disease 2019 (COVID-19) is a pandemic due to the novel severe acute respiratory virus-2 (SARS-CoV-2) [1] with more than 437 million cases and 6 million deaths in March 2022. Epidemiological studies have suggested that individuals with chronic underlying comorbidities such as hypertension, coronary artery disease, or diabetes are at risk for a severe outcome. Beyond the interest of vaccination, therapeutic strategies targeting the immune response or endothelial cell barrier integrity are currently under investigation in clinical trials. Viruses use the human cell environment for replication and induce abnormal function in host cells.

Several authors, considering: (1) the importance of intracellular calcium for the life cycle of viruses, (2) the positive association between hypocalcemia with COVID-19 severity; (3) the importance of promoting perfusion with pulmonary vasodilatation, have suggested an interest in calcium channel blockers (CCBs) for the treatment of people infected with COVID-19 [2,3,4]. To satisfy their need for calcium, several viruses induce an increased flux of calcium across cell membranes: CCBs reduce the levels of intracellular calcium [2]. CCBs, notably amlodipine, also have anti-inflammatory and anticoagulation effects in humans and cell models [2]. Several authors have proposed the hypothesis that hypocalcemia could be a physiological state characterizing COVID-19 and representing a beneficial host defense rather than a pathology [2]. The inhibitory effect of CCBs during viral infections has been reported for verapamil, an influenza A virus inhibitor, cilnidipine and manidipine, inhibitors for Japanese encephalitis virus, and L-type Ca _V_ channel inhibitor gabapentin, effective against hemorragic fever arenavirus [5].

There are two distinct chemical classes of CCBs: the dihydropyridines (DHP) (such as nifedipine and amlodipine) and the non-dihydropyridines (diltiazem and verapamil).

Our interest in CCBs is based on in vitro or clinical studies. Six in vitro studies [6,7,8,9,10,11] have reported that several CCBs (amlodipine, benidipine, cilnidipine, diltiazem, efonidipine, felodipine, fendiline, lercanidipine, manidipine, nicardipine, nifedipine, and verapamil) can inhibit SARS-CoV-2 replication (see Discussion for detailed descriptions of the studies).

Several retrospective or prospective clinical studies have explored the association between use of CCBs and clinical outcome in people infected with COVID-19. These studies have been analyzed in two meta-analyses [12,13]. In the first meta-analysis [12], 31 eligible studies were retained, including 119,298 patients with COVID-19 with or without hypertension, and CCB use was not associated with reduced mortality. Subgroup analysis in patients with hypertension (10 studies with 11,548 patients) revealed significantly reduced mortality (OR = 0.69; 95% CI: 0.52–0.91, *p* = 0.009). No difference in use of dihydropyridine was reported (5 studies with 5895 patients with or without hypertension). The second meta-analysis [13] explored the clinical outcome of only patients with hypertension infected with COVID-19. The authors retained 9 studies with 18,835 patients, and the meta-analysis of seven studies with 8413 patients revealed a significant reduction of mortality with the pre-admission use of CCBs (OR = 0.65; 95% CI 0.49–0.86). There was only one common study between the two meta-analyses.

Thus, the two meta-analyses reported that, in patients with hypertension infected with COVID-19, significant reduced mortality was associated with CCB use, confirming the in vitro studies that found significant effects of CCB drugs on SARS-CoV-2 replication. However, this relationship has not been explored in hypertensive patients when only dihydropyridine (DHP) drugs are prescribed. One open and unresolved question is the possibility that DHP drugs could have variable action on the SARS-CoV-2. For example, among the six in vitro studies, two [6,10] reported the superiority of amlodipine over other DHP drugs. The cause for the different effects of DHP drugs could be related to an additional mechanism of action against the SARS-CoV-2.

Acid sphingomyelinase (ASM) plays a crucial role in viral infection and the antiviral properties of functional inhibitors of ASM have been reported for several decades. Various FDA-approved drugs have been tested in vitro to explore their potential inhibition of ASM. Kornhuber et al. [14] have identified 72 FIASMAs that are defined by a reduction of ASM activity of at least 50% at 10 μΜ concentration. Among the FDA-approved drugs tested there were four DHP drugs (amlodipine, lercanidipine, barnidipine, cilnidipine) and only amlodipine had a reduction of ASM activity of at least 50% (88%). To the best of our knowledge, the FIASMA properties of the other DHP drugs have not been studied.

Several studies (see review in [15]) have found that FIASMAs have anti-SARS-CoV-2 properties, notably amlodipine. 

Among the different FIASMAs, selective serotonin reuptake inhibitors (SSRI), notably fluvoxamine and fluoxetine, are the most studied drugs. Several in vitro studies have reported that fluoxetine inhibited the entry and propagation of SARS-CoV-2 in the cell culture model without cytotoxic effects [16,17], and two studies have found that combination therapy including fluoxetine exerts antiviral effects against SARS-CoV-2. The first study [18] reported that a combined treatment with fluoxetine and GS-441524, the principal serum metabolite of remdesivir, displayed antiviral effects against three SARS-CoV-2 variants in vitro. The second study [19] tested the fluoxetine–remdesivir combination on the SARS-CoV-2 replication in vitro and found impaired viral replication with combination treatment to be well-tolerated. Several clinical retrospective and prospective studies have been reported, suggesting that fluvoxamine and fluoxetine could be a potential treatment of COVID-19. However, important methodological limitations of the different studies mean they should be interpreted with caution [20].

The studies on amlodipine comprised five in silico studies, three in vitro studies, and five clinical studies. There were four retrospective studies [6,21,22,23] and only one prospective study [24] that showed a more favorable outcome in four of the five studies for subjects treated with amlodipine compared to those not treated with amlodipine (see discussion for detailed descriptions of the studies). The effect of FIASMAs is related to the duration of the prescription within the study. The elimination half-life of amlodipine ranges from 30 to 50 h, therefore a duration of at least 9 to 15 days (7 half-lives) is required.

The characteristics of the clinical studies examining the relationship between amlodipine and mortality are summarized in Appendix A. Unfortunately, there are not any studies exploring the mortality rate only for inpatients with hypertension and COVID-19 infection, with chronic prescription of amlodipine, taking into account the comorbidities and confounding covariables.

In order to meet the limitations of the previous retrospective studies we have performed a new analysis of our retrospective cohort study [22].

## 2. Results

Of the participants, 184 with hypertension were retained (see Table 1). Of these 184 participants, 55 were treated with amlodipine (before the infection) and 129 received either antihypertensive drugs (including amlodipine prescribed after the hospitalization, and 22 patients receiving at least one another FIASMA) or had no antihypertensive drugs.

For the first analysis (55 Amlodipine versus 129 *n*-Amlodipine), five univariate logistic regressions reported significant associations between mortality and age, chronic kidney disease, malignant neoplasm, dementia, and amlodipine status. All five significant predictors were introduced in the multivariate logistic regression analysis. The regression was significant (Chi square (5) = 29.11; *p* < 0.0001). Chronic kidney disease and malignant neoplasm were significant predictors, the odds ratio with 95% confidence interval (95% CI) were, respectively, 2.16 (95% CI: 1.04–4.5; *p* = 0.039) and 2.46 (95% CI: 1.01–6.01; *p* = 0.047). For amlodipine status the odds ratio was 0.29 (95% CI: 0.11–0.74; *p* = 0.009) (see Table 2).

For the second analysis (55 Amlodipine versus 107 n-Amlodipine and n-other FIASMAs), five univariate logistic regressions reported significant associations between mortality and age, chronic kidney disease, malignant neoplasm, dementia, and amlodipine status. All five significant predictors were introduced in the multivariate logistic regression analysis. The regression was significant (Chi square (5) = 26.66; *p* < 0.0001). Chronic kidney disease and malignant neoplasm were significant predictors, as well as amlodipine status. The odds ratio with 95% confidence interval (95% CI) were, respectively, 2.47 (95% CI: 1.09–5.58; *p* = 0.028) and 3.11 (95% CI: 1.16–8.31; *p* = 0.022). For amlodipine status the odds ratio was 0.3 (95% CI: 0.11–0.78; *p* = 0.013) (see Table 2).

## 3. Discussion

The main result of the present study is a significant relationship between chronic intake of amlodipine and decreased mortality in inpatients with hypertension and COVID-19 infection. This significant relationship was found in unadjusted analyses as well as in adjusted analyses, taking into account sociodemographic variables and comorbidities. The results of the present study could suggest a protective effect of chronic intake of amlodipine in patients with COVID-19.

Six in vitro studies reported that several CCBs, notably amlodipine, could be candidates in the treatment of SARS-CoV-2.

The first study [6] tested whether CCBs can inhibit SARS-CoV-2 replication. Vero E6 cells were treated with a panel of nine CCBs (benidipine, amlodipine, cilnidipine, nicardipine, nifedipine, isradipine, nimodipine, nisodipine, felodipine) and then infected with SARS-CoV-2. Benidipine, amlodipine, cilnidipine, and nicardipine showed more significant effect, and antihypertensive drugs angiotensin II receptors blockers (ARBs) and angiotensine-concerting enzyme inhibitors (ACEIs) did not show an in vitro anti-SARS-CoV-2 effect.

The second study [7] screened 2000 approved drugs and found that 17 could be inhibitors of the main protease (Mpro) of SARS-CoV-2. These 17 drugs were tested in vitro (kinetic assay) for Mpro inhibition. Among the 17 drugs, 5 provided IC50 values below 40 μΜ: manidipine, boceprevir, lercanidipine, bedaquiline, and efonidipine.

The third study [8] screened 1700 USA FDA-approved using human coronavirus strain OC43 to identify drugs with anti(corona)viral effects. The retained compounds were screened for their inhibition of SARS-CoV-2 using Vero cells. Twenty anti-SARS-CoV-2 drugs were retained, including two CCBs: amlodipine and fendiline.

The fourth study [9] screened 1971 FDA-approved drugs using a nano-luciferase SARS-CoV-2 tag. Thirty-five drugs, including manidipine, reduced replication in vero cells and human hepatocytes when infected prior to SARS-CoV-2.

The fifth study [10] selected five drugs from different classes that all inhibit high voltage-activated Ca2+ channels of L-type (three dihydropyridines: amlodipine, nifedipine, felodipine; phenylalkylamine: verapamil; benzothiazepine: diltiazem). Vero E6 cells were infected with SARS-CoV-2. Amlodipine, felodipine, and nifedipine were highly selective against SARS-CoV-2.

The sixth study [11] identified eight small molecules, including amlodipine, capable of reversing the transcriptional landscape induced by SARS-CoV-2 infection. These eight drugs were tested on Vero-E6 cells infected with SARS-CoV-2 as well as on human pluripotent stem-cell-derived pancreatic endocrine organoid cultures. Among the eight drugs, four were the most potent (amlodipine, berbamine, loperamide, and terfenadine).

Thus, among the six studies that included at least one DHP drug, four included amlodipine. Among these four studies, three reported that amlodipine was one of the most potent drugs retained.

Five clinical studies have explored the association between amlodipine intake and prognosis in patients infected with COVID-19 (see Appendix A for studies exploring mortality).

The first retrospective study [21] studied 65 inpatients who were at or above the age of 65 years. Twenty-four patients took amlodipine and nifedipine and 41 did not take CCBs. The rates of hypertension were, respectively, 91.7% (*n*= 22) and 82.9% (*n*= 34): the difference was not significant. The mortality rates were, respectively, 50 and 85.4% in the CCB and non-CCB groups (*p* = 0.0036). No information on the duration of prescription before hospitalization was given.

The second study [6] explored 96 hospitalized patients with COVID-19 with hypertension (without other comorbidities): 19 were treated by amlodipine and 77 treated by non-amlodipine. The rates of mortality were, respectively, 0 (*n* = 0) and 19.5% (*n* = 15) on amlodipine and non-amlodipine (*p* = 0.0037). No information on the duration of prescription before hospitalization was given.

The third study [22] explored 317 hospitalized patients with COVID-19: 60 patients (55 with hypertension) on chronic intake of amlodipine and 257 (129 with hypertension) without chronic intake of amlodipine or other FIASMAs. Multiple regression controlling for age and comorbidities found that amlodipine intake was significantly associated with lower risk of intubation or mortality (Adjusted OR = 0.24, *p* = 0.0031).

The fourth study [23] reported on, in an observational multicenter study on individuals hospitalized for severe COVID-19, the relationship between FIASMAs and risk of intubation or death. There were 97 patients receiving amlodipine (% hypertension unknown) and these were compared to 2569 patients (873 with cardiovascular disorders, 34%) who had no FIASMA medication. Amlodipine was received within the first 24 h of hospital admission. Multivariable Cox regression analysis found that amlodipine intake was significantly associated with lower risk of intubation or mortality (Hazard Ratio = 0.7, *p* = 0.037).

The prospective study [24] was a randomized clinical trial comparing hospitalized primary patients with hypertension receiving either 50 mg of losartan or 5 mg of amlodipine per day for 2 weeks. The main outcomes were 30-day mortality rate and length of hospital stay. No significant difference between the two groups was reported. Unfortunately, the study has many weaknesses (e.g., no a priori calculation of number of subjects to detect significant difference, no a priori stratification on age before randomization), calling the results into question [25].

The strengths of the present study relative to other published studies (see Appendix A) are the hypothesis of a particular effect of amlodipine related to its FIASMA status, the chronic prescription of the drug, the mortality as the main criterion, the study on only inpatients with COVID-19 with hypertension, the control of other comorbidities, and the control of prescription of other FIASMAs. To the best of our knowledge no study has explored the role of amlodipine in patients with COVID-19 with hypertension using this methodological caution.

The limitations of the study are related to its retrospective design, the use of only one hospital for the recruitment, and that the precise cause of death has not been verified by systematic autopsies.

Concerning bioavailability, amlodipine has a low hepatic extraction ratio leading to a high bioavailability (around 65%) and is metabolized by CYP3A4 with a small contribution of CYP3A5. However, co-administration of strong CYP3A4 inhibitors may increase amlodipine plasma exposure. Co-administration amlodipine with indinavir co-administered with low-dose ritonavir (a potent CYP3A4 inhibitor) has led to a 1.9-fold increase in area under the curve (AUC) [26]. However, co-administration of amlodipine with grapefruit juice did not modify its AUC, while other dihydropyridines showed an increase in AUC (2- and 3-fold for nisoldipine and felodipine, respectively) [27]. Hence, co-administration with strong CYP3A4 inhibitors including drugs to be used in COVID-19 treatment such as Paxlovid (association of nirmatrelvir and ritonavir) may be used with caution, and dose adjustment strategies have been described [28]. Caution should also be considered in patients with HIV with ritonavir-boosted treatments, where halving the dosage has been suggested [29].

Concerning the mechanisms of action, amlodipine and other CCBs reduce levels of intracellular calcium, exert anti-inflammatory and anticoagulation effects, and have vasodilatory effects in the lungs and vascular system [2].

From a genetic point of view, one author has tested the hypothesis that the excessive immune-defense reactions with inflammation and coagulation (cytokines storm) could be a primary cause of death in patients with COVID-19 [30]. Genetic studies have explored the genetic risk factors associated with COVID-19 mortality. Allelic variation at seven single nucleotide polymorphisms (SNP) has been significantly associated with COVID-19 mortality. These seven SNPs were relevant to inflammation, coagulation, and respiratory functions. The association study exploring the phenotypes that could be associated to the seven SNPs has reported that use of amlodipine has been linked with four of the seven SNPs (see Table 1 in reference [30]).

In addition to the effect on calcium metabolism, amlodipine as a FIASMA has a protective ability against SARS-CoV-2, as shown in in vitro studies [15].

In addition to the effect on SARS-CoV-2, it is interesting to point out that both CCBs (verapamil, nimodipine, diltiazem) and fluoxetine reduced ebola infection efficiency using in vitro models [31,32].

## 4. Materials and Methods

### 4.1. Study Design and Participants

We used data from the Erasme Hospital of Brussels (Belgium) using the International Severe Acute Respiratory and Emerging Infections Consortium (ISARIC) COVID-19 database. The database includes individuals for whom data collection commenced on or before 1 September 2020. The study was approved by the Ethics Committee of the Erasme Hospital (Protocol P2020/358 approved on 16 July 2020). Using the initial cohort of 616 individuals, we selected 350 participants with SARS-CoV-2 infection confirmed by laboratory testing who were either alive on the 31 August 2020, or deceased in the hospital during the course of their hospitalization. Seventy-two patients died during hospitalization and 278 survived. Demographic data regarding symptoms, comorbidities, laboratory findings on admission, and outcomes were retrieved from the ISARIC COVID-19 database of the Erasme hospital and retrospectively reviewed and analyzed. The “Strengthening the Reporting of Cohort Studies in Epidemiology” statement guidelines were followed in the conduct and reporting of the study.

Patients with a long-term prescription of FIASMAs at admission were noted as FIASMAs positive (F+). Chronicity was defined as a duration equal to or higher than seven half-lives for each FIASMA (half-life range: 35–50 min (melatonin) to 20–100 days (amiodarone).

In Belgium, 28 FIASMAs are available, including one calcium-channel blocker (amlodipine), one Class 1 and 3 antiarrhythmic drugs (amiodarone), and one beta blocking agent (carvedilol). The participants who did not receive chronic FIASMAs on admission were noted as FIASMAs negative (F-). If a patient who did not receive chronic FIASMAs on admission had a new prescription of FIASMAs on the first day of hospitalization then they were classified as FIASMAs negative except for melatonin (7 half-lives: 6–7 h), alverine (7 half-life: 6 h), or cloperastine (7 half-lives: 24 h).

The following comorbidities were recorded: diabetes, hypertension, chronic liver disease, chronic rheumatic disease, chronic kidney disease, chronic cardiac disease (except hypertension), malignant neoplasm, chronic neurological disorders (except dementia), dementia, chronic lung disease (except asthma), chronic hematologic disease, asthma, obesity (defined by body mass index [BMI] > 30 kg/m^2^), and smoking.

In the previously published study the analysis was completed on 317 subjects (60 subjects on amlodipine and 257 subjects FIASMAs negative). For the present study only the 184 patients with hypertension were retained (see Appendix A flow chart and Appendix A). Fifty-five patients were treated with amlodipine (before the infection) and 129 received either antihypertensive drugs (including amlodipine prescribed after the hospitalization) or had no antihypertensive drugs. Among the 129 subjects, those receiving antihypertensive treatment had as the main pharmacological class of drugs: beta-blockers (32%), angiotensin-converting enzyme inhibitors (19%), angiotensin receptor blockers (16%), CCBs (16%), and diuretics (13%). Among the 129 subjects, 22 received at least one other FIASMA. The prescribed daily dose of amlodipine on the 55 subjects was given using the rate of defined daily dose (DDD). The rate of DDD for amlodipine was 137.6% (SD = 53%). Since the DDD of amlodipine is 5 mg per day, the rate of DDD for amlodipine on the 55 subjects was 6.88 mg per day (SD = 2.65). Antibiotics were received by 82.2% patients, and 10.5% received antivirals (notably remdesivir).

### 4.2. Statistical Analysis

Univariate and multivariate logistic regression models were used to evaluate risk factors of mortality among patients with COVID-19 using Statistica (version 7.1) software [33]. The dependent variable was the status (alive or deceased) and the independent variables were sex, age, each comorbidity, smoking, and amlodipine status.

From univariate analyses, each independent variable with significant association with mortality (*p* < 0.05) was included in the multivariate analysis, except for the amlodipine status, which is a forced variable.

Two analyses were completed.

First, the 55 subjects treated chronically by amlodipine were compared with the 129 patients treated or not for their hypertension (including patients receiving an acute prescription of amlodipine at the start of hospitalization). The mortality rates were, respectively, 12.73% (*n* = 7) and 34.88% (*n* = 45) for the amlodipine and non-amlodipine groups.

Second, the 55 subjects treated chronically by amlodipine were compared with the 107 patients treated or not for their hypertension and who did not receive another FIASMAs given chronically (*n* = 22), as these drugs could have protective effects on patients with COVID-19. The mortality rates were, respectively, 12.73% (*n* = 7) and 33.64% (*n* = 36) for the amlodipine and non-amlodipine groups.

## 5. Conclusions

The present study suggests a protective effect of amlodipine in inpatients with COVID-19 and hypertension. It must be confirmed, firstly, in larger multicenter retrospective studies, and secondly, by double-blind controlled randomized clinical trials in patients with COVID-19 and hypertension. Re-analysis of data on amlodipine of previously published CCB retrospective studies is also suggested. Taking into account that there is some data for, firstly, the combination treatment in cells infected by SARS-CoV-2 with remdesivir or its predominant serum metabolite GS-441524 with fluoxetine [18,19], or secondly, the combination treatment in infected cells with remdesivir and diltiazem and another CCB [34], combination treatment using amlodipine with remdesivir could be investigated.

From a practical point of view, we can suggest, only for patients with COVID-19 infection and treated by CCBs for their hypertension, a switch to amlodipine.

## Figures and Tables

**Table 1 pharmaceuticals-15-00380-t001:** Demographic and clinical characteristics of inpatients with COVID-19 with hypertension (*n* = 184).

Demographic/Clinical Characteristics.	*n (%)*
Age	69.54 +/− 14.6 *
Sex	
Female	81 (44)
Male	103 (56)
Comorbidities	
Diabetes	76 (41.3)
Chronic lung diseases	38 (20.6)
Chronic liver diseases	23 (12.5)
Chronic cardiac diseases	81 (44)
Chronic rheumatic disease	33 (17.9)
Chronic kidney disease	60 (32.6)
Malignant neoplasm	28 (15.2)
Chronic neurologic disorders	47 (25.5)
Dementia	20 (10.9)
Chronic hematologic disease	29 (15.8)
Asthma	13 (7)
Obesity	54 (29.3)
Smoking	13 (7.1)
Mortality	52 (28.3)
Amlodipine (chronic)	55 (29.9)
Other antihypertensive drugs	26 (14.1)
No antihypertensive drugs	103 (56)

* Mean +/− SD.

**Table 2 pharmaceuticals-15-00380-t002:** Risk factors for mortality among patients with COVID-19 with hypertension.

*n* = 184, (55 Amlodipine, 129 *n*-Amlodipine)							*n* = 162, (55 Amlodipine, 107 *n*-Amlodipine)					
Variable	Univariate Analysis			Multivariable Analysis			Univariate Analysis			Multivariable Analysis		
	OR	95% CI	*p*-Value	aOR	95% CI	*p*-Value	OR	95% CI	*p*-Value	aOR	95% CI	*p*-Value
Age	1.04	1.01–1.07	0.0018	1.03	0.99–1.06	0.05	1.03	1–1.06	0.016	1.02	0.98–1.05	0.31
Female sex (vs. male)			NS						NS			
Comorbidities												
Diabetes			NS						NS			
Chronic lung diseases			NS						NS			
Chronic liver diseases			NS						NS			
Chronic cardiac diseases			NS						NS			
Chronic rheumatic disease			NS						NS			
Chronic kidney disease	2.57	1.31–5.02	0.006	2.16	1.04–4.5	0.039	2.71	1.31–5.62	0.007	2.47	1.09–5.58	0.028
Malignant neoplasm	2.6	1.13–5.97	0.023	2.46	1.01–6.01	0.047	3.45	1.4–8.5	0.007	3.11	1.16–8.31	0.022
Chronic neurologic disorders			NS						NS			
Dementia	2.9	1.12–7.51	0.027	1.57	0.55–4.51	0.4	3.66	1.23–10.89	0.019	2.25	0.64–7.87	0.2
Chronic hematologic disease			NS						NS			
Asthma			NS						NS			
Obesity			NS						NS			
Smoking			NS						NS			
AMLODIPINE	0.27	0.11–0.65	0.003	0.29	0.11–0.74	0.009	0.29	0.12–0.70	0.006	0.3	0.11–0.78	0.013

OR: Odds ratio; aOR: adjusted odds ratio; 95% CI: 95% confidence interval. NS: no significance; 107 *n*-Amlodipine = 129 *n*-Amlodipine -22 subjects receiving another chronic FIASMA.

## Data Availability

Anonymized individual-level data from the survey will be made available through the International Severe Acute Respiratory and Emerging Infections Consortium (ISARIC) COVID-19 database. Requests for access will be reviewed by the data access committee of the Erasme hospital.

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
