# Peer review of "Association of Amlodipine with the Risk of In-Hospital Death in Patients with COVID-19 and Hypertension: A Reanalysis on 184 COVID-19 Patients with Hypertension"

_pharmaceuticals, 2022, doi:10.3390/ph15030380_

Round 1

Reviewer 1 Report

This MS presents new evidence that focuses on the effects of long-term use of amlodipine for hypertension on risk of Covid-19 mortality. The issue is very topical and important and the authors have conducted a nicely controlled study that shows strong effects of amlodipine in reducing nortality.  A number of issues can be addressed to improve the MS:

  • Lines 103-104 should read: ‘Unfortunately, there are ***not** any studies exploring the mortality rate …..
  • The start of the Results section should be made more clear by briefly explaining the context of the results presented in the first two paragraphs.
  • The authors should clarify the nature of the comparison between subjects taking amlodipine and other subjects. Were the comparison subjects taking other hypertension drugs? If so, which ones?
  • The authors should briefly discuss the expected implications of their results, with regard to reducing Covid-19 mortality. For example, should people with hypertension taking antihypertensive drugs consider switching to amlodipine, due to its positive effects against Covid-19?
  • The authors should spend a few more sentences explaining the relevance of the material in citation #2 to their study.
  • The authors should see https://academic.oup.com/emph/article/2020/1/314/5923290  Table 1, which shows that amlodipine use is associated with Covid-19 genetic risk factors, and briefly discuss this result.
  • The authors should briefly discuss what data should be collected next, and what analyses done, to further evaluate the use of amlodipine against Covid-19.

Author Response

Reponses to reviewer 1

  • The correction is done (see line 126)
  • A paragraph is added explaining the context of the results (see lines 133-136)
  • The nature of the comparison is clarified mentioning (lines 317-319) the different classes of antihypertensive agents (see lines 317-320)
  • See lines 354-355 concerning the pratical implication of the results
  • The reference 2 (Crespi) is developed (see lines 58-60)
  • The reference 30 is presented and developed (see lines 266-274)
  • To evaluate the interest of amlodipine in COVID-19 several ways of research are presented (see lines 349-353)

Reviewer 2 Report

In the manuscript entitled "Association of amlodipine with the risk of in-hospital death in patients with COVID-19 and hypertension: a reanalysis on 184 Covid-19 patients with hypertension”, the authors analyzed the effect of amlodipine in COVID-19 patients with hypertension. A broad range of studies has already confirmed that hypertension is a major disease that increases the risk of acute respiratory failure, hospital admission and mortality rate among patients with COVID-19. The authors analyzed the effect of amlodipine, which is both a CCB and a FIASMA, on the disease outcome. Many reports have demonstrated that FIASMAs e.g. fluoxetine or fluvoxamine significantly impairs viral infection in vitro or in clinical patients cohorts. Thus, the authors’ initiative to provide analyzed clinical data on patients treated with amlodipine and that were suffering from hypertension is welcome.

The overall merit of the here presented manuscript is high and clearly demonstrate the potential of drugs that either target calcium channel blocker, acid sphingomyelinase or both to ameliorate COVID-19 in patients with hypertension. The study is a valuable addition to the following study (PMID: 33650197). However, I have some concerns that the authors need to address before the manuscript can be considered for publication.

  1. Could the authors add information on the daily drug dose given to the patients, this would provide a valuable input for future studies.
  2. What is the bioavailability of amlodipine in patients? (PMID: 29427135)
  3. Did any of the patients received a treatment with antivirals e.g. remdesivir?
  4. While the clinical cohort design is correct and including the effect of other FIASMAs in patients (these patients were excluded in the study), I would like to encourage the authors to elaborate the introduction and discussion section about the direct antiviral effect of FIASMAs and CCBs on SARS-CoV-2 infections. The following reference are helpful for elaborating the introduction and discussion and should be included (PMID: 33732462, PMID: 35053369, PMID: 33650197, PMID: 32975484, PMID: 33572117, PMID: 33825201, PMID: 34575474)
  5. Could the authors discuss a possible combinatory treatment of amlodipine with direct antiviral agents (DAA) e.g. remdesivir. There are still some data for the combinatory treatment in cells with remdesivir or its derivatives GS-441524 with fluoxetine available (PMID: 33825201, PMID: 34575474). What is the authors’ opinion to combine amlodipine with remdesivir?
  6. The authors should also discuss if amlodipine is suitable for the treatment of other virus infections. Recently, it was shown that the FIASMA fluoxetine impairs Ebola virus infection (PMID: 34919035).

I hope that the authors can provide a revised version of their manuscript (more information about the study design, see above) and with an elaborated introduction and discussion section.

Author Response

Reponses to reviewer 2

  • The daily dose of amlodipine is given (see lines 320-325)
  • Bioavailability of amlodipine is presented in the discussion section (see lines 248-260) with 4 news references.
  • The rate of patients receiving antibiotics or antiviral sis mentioned (see lines 324-325)
  • The effects of FIASMAs and CCBs on SARS-CoV-2 are presented in Introduction and Discussion. As suggested by the reviewer 6 new references are presented .
  • The combinary treatment of amlodipine with remdesivir or its principal metabolite is discussed (see lines 349-353) and two references are presented in the introduction (see lines 108-113)
  • The interest of CCBS and notably amlodipine for the treatment of other virus infection is presented in the Introduction (see lines 60-63) and in the Discussion (see lines 277-279). Two references were added.

Round 2

Reviewer 2 Report

The authors have addressed all my comments and elaborated the introduction and discussion section as suggested. The revised manuscripts meets all the criteria for publication.